# Evolution of Glutamate Metabolism via *GLUD2* Enhances Lactate-Dependent Synaptic Plasticity and Complex Cognition

**DOI:** 10.3390/ijms25105297

**Published:** 2024-05-13

**Authors:** Andreas Plaitakis, Kyriaki Sidiropoulou, Dimitra Kotzamani, Ionela Litso, Ioannis Zaganas, Cleanthe Spanaki

**Affiliations:** 1Department of Neurology, School of Health Sciences, Faculty of Medicine, University of Crete, Voutes, 71003 Heraklion, Crete, Greece; dkotzamani@yahoo.gr (D.K.); irenelts20@gmail.com (I.L.); johnzag@yahoo.com (I.Z.); 2Department of Biology, University of Crete, Voutes, 71003 Heraklion, Crete, Greece; sidiropouloukiki@gmail.com; 3Institute of Molecular Biology and Biotechnology, Foundation for Research and Technology Hellas (IMBB-FORTH), 70013 Heraklion, Crete, Greece; 4Neurology Department, PaGNI University General Hospital of Heraklion, 71500 Heraklion, Crete, Greece

**Keywords:** *GLUD2*, CA1/CA3 LTP, lactate, glutamate, synaptic plasticity, human brain evolution

## Abstract

Human evolution is characterized by rapid brain enlargement and the emergence of unique cognitive abilities. Besides its distinctive cytoarchitectural organization and extensive inter-neuronal connectivity, the human brain is also defined by high rates of synaptic, mainly glutamatergic, transmission, and energy utilization. While these adaptations’ origins remain elusive, evolutionary changes occurred in synaptic glutamate metabolism in the common ancestor of humans and apes via the emergence of *GLUD2*, a gene encoding the human glutamate dehydrogenase 2 (hGDH2) isoenzyme. Driven by positive selection, hGDH2 became adapted to function upon intense excitatory firing, a process central to the long-term strengthening of synaptic connections. It also gained expression in brain astrocytes and cortical pyramidal neurons, including the CA1-CA3 hippocampal cells, neurons crucial to cognition. In mice transgenic for *GLUD2*, theta-burst-evoked long-term potentiation (LTP) is markedly enhanced in hippocampal CA3-CA1 synapses, with patch-clamp recordings from CA1 pyramidal neurons revealing increased sNMDA receptor currents. D-lactate blocked LTP enhancement, implying that glutamate metabolism via hGDH2 potentiates L-lactate-dependent glia–neuron interaction, a process essential to memory consolidation. The transgenic (Tg) mice exhibited increased dendritic spine density/synaptogenesis in the hippocampus and improved complex cognitive functions. Hence, enhancement of neuron–glia communication, via *GLUD2* evolution, likely contributed to human cognitive advancement by potentiating synaptic plasticity and inter-neuronal connectivity.

## 1. Introduction

The mammalian brain is a highly complex organ comprising diverse types of cells supporting a multitude of functions essential for the survival and well-being of the organism. These include homeostasis control, motor activities, eye movements, sensory perception, innate behaviors, and cognition [1]. Humans also possess unique cognitive abilities, including language function and symbolic thought. Further, humans are set apart by their “social cognition” and “cumulative culture” [2]. While it is well-known that the human brain displays discrete regions serving specific functions, the underlying molecular genetic mechanism(s) driving this specialization and leading to unique human capabilities are still poorly understood [1,3].

The human brain is distinguished from that of apes by its large size (mainly neocortical expansion) and unique architectural organization, a feature largely reflecting evolutionary changes in cortical neurons and glial cells and their connectivity. Specifically, there is an increased proportion and diversity of long-range projecting neurons of the human cortex associated with increased cortico-cortical connectivity, particularly in cortical areas involved in language function [4]. Also, the excitatory pyramidal neurons of the human cerebral cortex, cells crucial to cognitive processes, exhibit increased dendritic branching and synaptic spine density [5,6]. Parallel evolutionary changes have also occurred in human brain glial cells, including astrocytes, oligodendrocytes, and microglia [4,7], with the evolution having targeted pathways related to neuron–glia communication [8].

These observations are congruent with the results of high-throughput single-cell profiling of the transcriptome and proteome [3], revealing that evolution did not introduce new cell types in the human brain. Instead, it induced variation in existing cells and changes in their distribution, in order to create distinct circuitry [3]. Indeed, comparative studies have shown that inter-neuronal connections are denser in the human brain than in the chimpanzee brain, particularly in the middle temporal gyrus [8]. Compared to inhibitory neurons, the excitatory projection neurons exhibit much greater regional differences in their proportion, distribution, and gene expression [9]. 

Besides its unique cytoarchitecture, the human brain is also defined by high levels of synaptic (mainly glutamatergic) activity and energy utilization [10,11]. The latter is due to high energy demands arising from intense excitatory transmission, a process essential for cognitive functions. Indeed, early efforts to understand the molecular and cellular basis of cognition revealed that the high-frequency excitatory drive of hippocampal synapses (conventionally produced in the laboratory by theta-burst electrical stimulation of CA1/CA3 synapses) results in long-lasting strengthening of synaptic responses or long-term potentiation (LTP) [12]. This activity-shaped modification of synapses, known as synaptic plasticity, brought about in vivo by enhanced glutamatergic firing, ultimately leads to synaptic remodeling or structural plasticity, including the formation of novel synaptic contacts (synaptogenesis) and the growth of dendritic spines [13]. As such, excitatory transmission plays a crucial role in the genesis of novel neuronal connections (microcircuitry) that represent the structural basis for long-term memory. 

The evolutionary origin of these adaptations remains elusive, given that genes encoding proteins involved in synaptic transmission and energy metabolism did not undergo significant evolutionary changes in the human lineage [14]. Consistent with this view are electrophysiologic data obtained in human and rodent brains showing that the fundamental unit of synaptic transmission is remarkably conserved in mammals [15]. Instead, evolutionary changes have occurred in the regulatory elements of genes expressed in the human brain [2,4,16,17], with more than 80% of adaptive sequence evolution in humans thought to be regulatory [8]. Such evolutionary adaptation has led to upregulation of the glutamatergic signaling pathway [18] and energy metabolism [10,19] in the human brain.

To this day, only a few new genes, thought to have played a role in human brain evolution, have been characterized [4]. Of these, *GLUD2* is of particular importance, given that (a) it emerged in the hominoid ancestor and evolved under positive selection concomitantly with brain development; (b) it encodes human glutamate dehydrogenase 2 (hGDH2), an enzyme central to the metabolism of glutamate, the major excitatory transmitter involved in cognitive processes, and (c) hGDH2 acquired unique functional properties that allow the novel enzyme to be called into action upon intense excitatory firing, a process required for long-term strengthening of synapses and the creation of inter-neuronal connections. The possibility that hGDH2 has contributed to the acquisition of traits unique to humans has been supported by recent investigations in mice transgenic for the human *GLUD2*, showing that the human gene enhances synaptic plasticity/synaptogenesis and complex cognition [20]. Moreover, enhancement of synaptic plasticity by *GLUD2* is lactate-mediated, thus providing additional evidence that synaptic lactate mechanisms are essential to memory consolidation. *GLUD2* has also adapted to the particular metabolic needs of non-neural tissues where expressed. This review aims to detail these advances and their implications for understanding the role of *GLUD2* in human biology.

## 2. Emergence and Evolution of the *GLUD2* Gene

While synaptic transmission has been conserved in the mammalian brain, evolutionary changes occurred in synaptic glutamate metabolism in the common ancestor of humans and apes (about 23 million years ago) with the emergence of the *GLUD2* gene [21,22]. The new gene arose through retro-transportation of a processed *GLUD1* mRNA to the X chromosome [21]. Such gene duplication is thought to advance evolution by generating genes with new functions [4,23]. While most new genes derive from large genomic segment reduplications [4], those generated via retro-transportation end up mostly as pseudogenes [14]. However, *GLUD2* avoided this fate by acquiring, soon after its retro-position to the X chromosome, new functions suited for the particular metabolic needs of primate tissues where expressed [24,25,26].

Following its emergence in the hominoid, *GLUD2* evolved in the human and ape lineages. As a result, the encoded human GDH2 acquired 15 evolutionary amino acid changes [21]. These conferred unique properties to hGDH2 that permit enzyme function under conditions inhibitory to its ancestor, the human glutamate dehydrogenase 1 (hGDH1) isoenzyme [24,25,26]. Studies, using site-directed mutagenesis of the *GLUD1* gene at sites that differ from the corresponding residues of *GLUD2*, have elucidated the molecular mechanisms by which these amino acid replacements provided hGDH2 with unique properties [24,25,26,27,28,29,30,31,32]. They revealed that two evolutionary amino acid substitutions (Arg443Ser, Gly456Ala) were largely responsible for the major functional differences between hGDH2 and hGDH1 [27,28], whereas other replacements provided more refined properties [24,25,26].

Phylogenic evidence suggests that the Arg443Ser and Gly456Ala changes occurred along with five additional evolutionary replacements (Ala3Val, Glu34Lys, Asp142Glu, Ser 174Asn, Asn498Ser) in the first few million years after the gene reduplication event (Figure 1; branch A–B). Six of these seven amino acid changes are present in every member of the hominoid radiation that possesses the *GLUD2* gene (Figure 1). An exception is the Hylobates moloch genus of the gibbon family, in which the Arg443Ser and Gly456Ala changes were reversed in association with the appearance of four new mutations [33]. These observations have raised questions regarding the positive selection of these sites during the *GLUD2* evolution, and the role of these residues in the functional adaptation of GDH2 [33]. Concerning the latter, it is presently unclear whether the four new amino acid replacements that emerged along with this reversal in Hylobates moloch provide to the gibbon GDH2 properties similar to those conferred by the Arg443Ser and Gly456Ala replacements to human GDH2 [33]. Moreover, particularly intriguing is the absence of gibbon *GLUD1* sequences in UniProt [33], raising the theoretical possibility (according to Aleshina and Aleshin) that *GLUD1* might have become a pseudogene after the emergence of *GLUD2* in the gibbon. While this seems unlikely, reversal of the Arg443Ser and Gly457Ala mutations reinstates ancestral amino acids present in hGDH1 that contribute to the housekeeping properties of the enzyme. As such, whether this reversal permitted the gibbon GDH2 to take over some of the metabolic duties of hGDH1 and whether this has affected the *GLUD1* evolution remain to be further explored. Additional studies, including the functional characterization of gibbon GDH2s, are needed to better understand not only the evolutionary trajectory of the *GLUD2* gene but also its impact on the biology of these species [33]. 

In addition to these changes in the mature GDH2, evolutionary replacements also occurred in the mitochondrial targeting sequence (MTS) of the protein. These conferred an enhanced mitochondrial targeting capacity [32,40]. Two positive selected evolutionary amino acid substitutions that occurred in the hominoid are thought to provide this novel property to GDH2. Specifically, the Glu7Lys evolutionary change, which is conserved among apes and which replaces a negatively charged residue (Glu) with a positively charged one (Lys), may play a key role in enhancing the transport of the GDH2 protein into the mitochondria [32]. Also, the Asp25His change, which replaces a negatively charged residue (Asp) with a partially positively charged residue (His), may have also contributed to the mitochondrial targeting adaptation of hGDH2 [32]. In the gibbon lineage, however, a three-amino-acid deletion (residues: 24, 25, and 26) that includes the Asp25His residue reduces the MTS enhanced targeting capacity [32]. (The residue numbering for the cleavable presequence (MTS) starts with Met1, whereas that for the mature GDH2 starts with Ser1).

The emergence of *GLUD2* and its subsequent evolution coincided with a period of increasing brain size and complexity and, as such, the novel gene may have advanced primate brain development [22]. However, because *GLUD2* is present in all members of the ape radiation and because no functional studies are available for non-human ape hGDH2, the contribution of the novel gene to the divergence of the human brain is unclear. On the other hand, the primary structure of human GDH2 differs substantially from that of the non-human apes. Thus, as shown in Figure 1, the human GDH2 differs from the chimpanzee GDH2 by 4 amino acid residues, from the gorilla GDH2 by 6, from the orangutan GDH2 by 10, and from the gibbon GDH2 by 12. These amino acid differences correlate significantly (Pearson correlation *r* = 0.9957; *p* = 0.004) with the time distances (in Mya) between the human and each ape and their last common ancestor (Figure 1).

Phylogenetic evidence also suggests that the human neocortex expanded rather rapidly about −2 Mya [41], a period which falls well after the split of the human and the chimpanzee lineages (6–8 Mya). However, the contribution of the *GLUD2* gene to this evolutionary leap remains to be better understood. Thus, because the functional characteristics of the chimpanzee GDH2 are not currently available, the potential impact of the four amino acid residues that distinguish human hGDH2 from chimpanzee GDH2 on human brain evolution remains unclear. Nevertheless, observations on transgenic mice carrying the human *GLUD2* gene have revealed an expression trajectory for *GLUD2* that is greatly similar to that observed during prefrontal cortex development in humans [42].

## 3. Functional and Structural Aspects of hGDH1 and hGDH2

The recent genesis of *GLUD2* enriched the human genome with two GDH-specific genes: the original *GLUD1* gene, common to all mammals, encoding the hGDH1 enzyme, and the duplicated *GLUD2* gene encoding the hGDH2 isoenzyme. While hGDH1 has been extensively studied over the past decades, hGDH2 has been the subject of more recent investigations [24]. As noted above, structure/function relationships in hGDH1 and hGDH2 have been determined based on mutagenesis studies [24,25,26,27,28,29,30,31,32] and structural analyses of the crystallized proteins [43,44]. Results of these investigations have advanced our understanding of the role of these proteins in human biology. 

As noted above, human GDH2 acquired 15 evolutionary amino acid changes [21] that provided unique properties to the novel enzyme [24,26]. In addition, under the influence of a different promoter, encountered in the X chromosome, *GLUD2* diversified its expressional profile [45,46,47]. Although hGDH1 has maintained its housekeeping role, being expressed widely (with the highest levels found in the liver), hGDH2 exhibits a distinct expression profile in human tissues. Specifically, hGDH2 is mainly expressed in the human brain, testis, kidney, and steroidogenic organs, but it shows little expression in the human liver [45,46,47]. In the human cerebral cortex, *GLUD2* is expressed both in astrocytes and in large neurons of pyramidal morphology [46]. Of particular importance are observations in mice carrying the human *GLUD2* gene, which revealed the expression of hGDH2 in the CA1-CA3 pyramidal cells of the hippocampus [20], neurons that play a crucial role in synaptic plasticity and cognitive processes. 

hGDH1 is an allosterically regulated enzyme, with GTP and ADP acting as the main endogenous negative and positive regulators, respectively [24,26]. While GTP potently inhibits (IC_50_ 0.1–0.2 μM) hGDH1, ADP activates the enzyme (AD_50_~18–20 μM), with the two allosteric effectors acting antagonistically. L-leucine also activates hGDH1, but at relatively high levels (5–10 mM) [26]. However, lower L-leucine concentrations sensitize hGDH1 to the stimulatory effect of ADP [26]. In addition to these regulators, palmityl-CoA, spermidine, and steroid hormones can also serve as endogenous enzyme effectors [24]. Also, binding to short-chain 3-hydroxyacyl-CoA dehydrogenase [48] and ADP ribosylation (of a cysteine) can reversibly inhibit hGDH1 activity [49]. Lastly, multiple exogenous compounds of diverse structures are also shown to influence GDH activity [24] but their role in biology is unclear. 

Given these complexities, understanding the physiological function of hGDH1 in mammals remains challenging. Nevertheless, oxidative deamination of glutamate by hGDH1 produces α-ketoglutarate, ammonia, and reducing equivalents (NAD(P)H). Subsequently, α-ketoglutarate enters the TCA cycle being decarboxylated oxidatively (via α-ketoglutarate dehydrogenase) to succinyl-CoA/succinate, a TCA cycle step that generates GTP from GDP. In turn, GTP, acting as an energy sensor, powerfully inhibits hGDH1, thus preventing glutamate from fueling the TCA cycle when ample cellular energy supplies (high ATP levels) prevail. However, under conditions of limited Acetyl-CoA availability, GTP synthesis via the TCA cycle is decreased, permitting enzyme activation by rising ADP concentrations. The resulting increased flux of glutamate through the hGDH1 pathway provides the α-ketoglutarate needed for sustaining TCA cycle function.

In contrast to hGDH1, hGDH2 dissociated its function from GTP control due to the Gly456Ala evolutionary replacement [27], which occurred shortly after the cDNA insertion into the X chromosome as noted above [22,34]. The new property allows enzyme function even when an active TCA cycle generates GTP concentrations inhibitory to hGDH1. The second important adaptation enabled hGDH2 to drastically reduce its basal catalytic activity (to about 4–6% of maximal), while remaining remarkably responsive to activation by ADP. (by ~2500% at 1.0 mM) and L-leucine (by ~1400% at 10.0 mM) [28]. This property is largely conveyed by the Arg443Ser replacement [28], which also occurred at the first evolutionary step (Figure 1, branch A–B). However, because the introduction of the Arg443Ser change renders the enzyme essentially inactive, other evolutionary amino acid replacements act in concert with the Arg443Ser change to provide hGDH2 with its unique properties [24,25,26]. The novel functional characteristics, acquired by hGDH2, permit specialized cells to utilize the hGDH2-catalyzed reaction to accomplish some of their unique functions as detailed below. 

At the structural level, mammalian GDH1 is a homo-hexamer comprising two trimers (Figure 2 presents the structure of hGDH2). Each of the six subunits encompasses the NAD+-binding domain, the glutamate-binding domain, the “pivot helix”, and the “antenna” (Figure 2). The latter is a 48-amino-acid protruding structure bearing a small C-terminal α-helix that undergoes prominent conformational changes during catalysis [43]. In the trimeric structure, the antennas of the adjacent subunits are intertwined, with this inter-subunit interaction playing an important role in setting basal catalytic activity and regulation [50,51]. Several mutations in hGDH1 that attenuate GTP inhibition leading to the hyperinsulinemia/hyperammonemia (HI/HA) syndrome are located in the antenna and in the pivot helix [52,53]. 

hGDH2 was also recently crystallized and its structure was determined [44]. These studies revealed that hGDH2 is also a hexamer composed of two trimers (Figure 2). Similarly to hGDH1, each monomer displays the NAD+ domain, the glutamate-binding domain, the pivot helix, and the antenna. The role of the pivot helix and the antenna in hGDH2 evolution is underscored by observations showing that evolutionary replacements that have profound functional consequences are located in these structures (Figure 2). Specifically, the Arg443Ser change is found in the small C-terminal α-helix of the antenna, while the Gly456Ala replacement is located in the pivot helix. Moreover, a study of the crystal structure of hGDH2 [44] has revealed that the enzyme adopts a novel semi-closed conformation, which may explain some of its unique functional properties, including its ability to remain dormant under baseline conditions and be called into action by raising the levels of ADP and L-leucine, as described above. 

Additionally, the crystallization of hGDH2 allowed in silico studies on the structural evolution of the primate protein [34]. These studies, using AlphaFold, examined changes in the GDH2 structure occurring during the evolutionary transition from extinct primate ancestors to modern apes, including humans. They revealed that the initial seven evolutionary amino acids, which occurred shortly after the retro-transposition event as detailed above (branch A,B in Figure 1), served as a basis for subsequent modifications that fine-tuned its enzymatic properties. 

While site-directed mutagenesis of the *GLUD1* gene at sites that differ from the corresponding residues of *GLUD2* proved crucial to identifying the evolutionary amino acid replacements that equipped hGDH2 with unique properties, mutagenesis studies performed on the *GLUD2* gene yielded interesting results [54]. Thus, while amino acid replacements in the pivot helix (shown in Figure 2) diminish enzyme activity and abrogate regulation, those located in the antenna (shown in Figure 2) increase enzyme activity without affecting regulation [54]. Consistent with this pattern is the finding that a rare naturally occurring polymorphism (Ala445Ser), located in the small C-terminal α-helix of the antenna (Figure 2), provides gain-of-function properties to hGDH2 [55]. Indeed, functional analyses of Ser445-hGDH2 revealed that the variant displays enhanced basal activity that is resistant to GTP control but is markedly sensitive to inhibition by estrogens [55]. Importantly, the hGDH2 variant was shown to accelerate Parkinson’s disease (PD) onset in hemizygous males, but not in heterozygous females [55]. The protection of female PD patients has been attributed to the modification of the hyperactive enzyme by estrogens [55]. While the mechanisms by which this hGDH2 variant speeds up PD progression remain to be better understood, enhanced oxidative deamination of glutamate by the hyperactive enzyme in the mitochondria may increase oxidative stress [55]. Zhang et al. [56] recently tested the effect of the Ser445Ala mutation in their PD model and found that introduction of the Ala445-hGDH2 variant exacerbated the movement abnormalities of the animals and the degeneration of the nigral dopaminergic neurons, probably by damaging the mitochondria of these cells [56].

## 4. Insights from the Study of Transgenic Mice Carrying the Human *GLUD2* Gene 

Recently, the use of a transgenic animal model, created by the insertion of a region of the human X chromosome containing the *GLUD2* gene and its regulatory elements, has yielded important insights into the role of hGDH2 in human biology [20,42,47]. Driven by its promoter in the human X chromosome, hGDH2 is expressed in the host mouse tissues in a pattern that is very similar to that observed in human tissues [47]. This includes expression in the brain, kidneys, pancreas, steroidogenic organs, and testis while showing little expression in the liver [47]. In contrast, the endogenous mouse GDH1 (mGDH1) is widely expressed, with the highest expression occurring in the liver. Also, the cellular expression pattern of hGDH1 and hGDH2 in Tg mouse tissues is essentially the same as that seen in human tissues [47].

## 5. The Role of hGDH1/2 in Cellular Energetics: Clues from the Regulation of Glucose Homeostasis

Given the regulatory pattern of hGDH1, as has been observed in vitro using purified enzyme preparations, and the fact that α-ketoglutarate, generated by oxidative deamination of glutamate, enters the TCA cycle leading to the synthesis of ATP, the hypothesis was advanced over a half a century ago that the enzyme is controlled by the cell’s energy needs [57]. Indeed, observations in patients harboring mutations in hGDH1 that attenuate GTP inhibition are consistent with this hypothesis [52,53]; they reveal that the hyperactive enzyme affects glucose homeostasis by boosting the cellular energy charge and consequently the insulin release from pancreatic β-cells. Such patients, known to suffer from the HI/HA syndrome (noted above), experience bouts of protein-diet-induced hypoglycemia, seizures, and other neurologic symptoms [58] attributed to untoward activation of hGDH1 in the β-cells. Specifically, L-leucine contained in the diet, acting synergistically with the endogenous ADP in β-cells, is thought to counteract the weak inhibitory effect of GTP on the mutant hGDH1, resulting in enzyme activation. Although additional mechanisms may be operational [59], increased glutamate flux through the GDH-TCA cycle pathway leads to decreased conductance of the ATP-dependent K^+^ channel resulting in insulin release.

## 6. The Potential Role of hGDH2 in Lactate Metabolism: Clues from Studies on Testicular Tissue

The original cloning of *GLUD2* led to the realization that the novel human gene is expressed in neural and testicular tissues [21]. Subsequently, immunofluorescence (IF)-confocal microscopy studies, using antibodies specific for hGDH1 and hGDH2, have established that hGDH2 is densely expressed by the Sertoli cells of the human testis [45]. In contrast, these cells do not express hGDH1. However, both hGDH1 and hGDH2 are expressed in the interstitial Leydig cells known to serve endocrine functions. An identical expression pattern has been more recently observed in the testis of mice transgenic for the human *GLUD2* gene [47]. These observations are consistent with single-cell RNA expression data showing high *GLUD2* expression levels in Sertoli cells [60,61]. As such, the dense expression of hGDH2 by Sertoli cells raises important questions regarding the role of the novel enzyme in human testis biology. 

The Sertoli cells, located in the seminiferous tubules, nourish germ cells (spermatocytes and spermatids) by providing them with lactate and other nutrients. They express the MCT systems for transporting lactate and other monocarboxylates across their membranes [62]. It is also well-known that germ cells utilize lactate, rather than glucose, as a preferred energy substrate for maintaining the exceptionally high ATP levels needed for sperm motility and other functions [63]. The importance of these mechanisms in germ cell biology is underscored by observations showing that disruption of Sertoli cell function results in reduced lactate production and spermatogenic failure [64].

Notably, the supporting role of Sertoli cells for germ cells is quite analogous to that of astrocytes for neurons [64]. Indeed, astrocytes interact metabolically with neurons by providing them with lactate to be used as a preferred energy substrate and for other functions [65]. While lactate derives mainly from glycogenolysis [63,65], glutamate metabolism through the GDH-TCA cycle also generates lactate (Figure 3) that is released by the cell [66,67,68]. In light of these considerations, the finding that Sertoli cells densely express hGDH2 suggests that glutamate metabolism represents another pathway for lactate production that operates in parallel with glycolysis. Hence, hGDH2 evolution has enhanced the ability of astrocytes and Sertoli cells to support neurons and germ cells, respectively, by supplying them with lactate. It is also of particular interest that the function of the supported cells (neurons and germ cells) requires very high energy consumption [10,63,65].

## 7. The Potential Role of GDH1/2 in Ammonia Metabolism: Clues from Studies on Renal Tissue

In human kidneys, hGDH2 is co-expressed with hGDH1 in the epithelial cells of the proximal convoluted tubules [24]. Similar results have been obtained in mice transgenic for *GLUD2* [47]. They are also confirmed by single-cell RNA expression analyses showing high *GLUD2* expression levels in the human renal tubular epithelium [60,61]. Due to its lower optimal pH [26], hGDH2 may operate more efficiently than hGDH1 during acidosis, probably helping the epithelial cells of the proximal convoluted tubules to excrete excess protons in the form of ammonium. It is known that the initial step in ammoniagenesis during acidosis is the deamination of glutamine to glutamate via the mitochondrial phosphate-dependent glutaminase (PDG). However, because the formed glutamate inhibits PDG, GDH steps in to remove the excess glutamate from the mitochondria. Indeed, acidosis is shown to significantly increase GDH activity in the proximal convoluted tubules [69]. Also, in patients with the HI/HA syndrome, the hyperactive mutant hGDH1 enhances the renal production of ammonium, possibly accounting for the hyperammonemia observed in this syndrome [70]. There is also evidence that the GDH function is crucial to renal processes requiring high energy utilization [71]. Hence, the co-expression of hGDH1 and hGDH2 in human renal tubules’ epithelium may provide a biological advantage to humans that needs to be further understood. 

Owing to their low affinity for ammonia (Km 15–30 mM) and the very low ammonia levels present in human tissues (about 30 μM) [59], hGDH1 and hGDH2 operate predominantly towards the oxidative deamination direction. However, high concentrations of ammonia (hyperammonemia) stimulate the reductive amination of α-ketoglutarate to glutamate in the nerve tissue [72]. While this is thought to be the function of the housekeeping hGDH1, recent studies on human brain astrocytes [73] have demonstrated that during hyperammonemia, hGDH2 removes ammonia by fixing it on α-ketoglutarate, a process that may inhibit the TCA cycle by consuming α-ketoglutarate. 

## 8. *GLUD2* in Brain Biology

### 8.1. Studies on Human Central Nervous Tissue

Regarding the role of the GDH pathway in the nervous system, morphological studies on the human brain have yielded important insights into the potential function(s) of hGDH1 and hGDH2 in this organ. While hGDH1 is only expressed in glial cells (including astrocytes, oligodendrocytes, and oligodendrocyte precursors), hGDH2 is expressed both in astrocytes and neurons [46]. In astrocytes, hGDH1 is expressed in mitochondria distributed throughout the cell body and in the astrocytic processes. On the other hand, in oligodendrocytes and their precursors, hGDH1 is expressed in the cell nucleus [46,47]. While this nuclear localization may appear aberrant, α-ketoglutarate dehydrogenase, another metabolic enzyme of the TCA cycle linked to GDH function, can also localize to the cell nucleus, where it generates α-ketoglutarate needed for gene regulation via histone modification [74]. Whether hGDH1 serves a similar function in dividing glial cells needs to be further understood. 

As noted above, hGDH2 in the human brain is expressed in neuronal and glial cells. Indeed, morphological studies using an anti-hGDH2-specific antibody have revealed that the novel enzyme is expressed in in GFAP-labeled astrocytes and in some cortical neurons characterized by a pyramidal morphology [46]. It has also been detected in the nuclear membrane of small cortical neurons [46,47]. In pyramidal neurons, hGDH2 localizes to mitochondria distributed both in the perikaryon and the cytoplasmic membrane in juxtaposition with astrocytic feet engulfing synapses [46,47]. Expression of *GLUD2* in excitatory neurons has been confirmed by single-cell RNA expression analyses [33]. These findings are consistent with the view that hGDH2 is involved in excitatory transmission mechanisms, an expectation that has received substantial support from recent electrophysiologic investigations in *GLUD2* transgenic mice, as described below. 

### 8.2. Studies on Tg Mice’s Central Nervous System 

Studies of the Tg mouse brain, using IF/confocal microscopy and antibodies specific for hGDH2, generated data that essentially replicated those obtained in the human brain. Briefly, these studies revealed that hGDH2 is expressed in the neuropil, where it localizes to GFAP-positive astrocytes. In addition, hGDH2 is densely expressed by some cortical neurons of the pyramidal morphology [47], in a pattern that is strikingly similar to that observed in the human brain [46]. In the hippocampus of the Tg mice, hGDH2 is expressed in CA1-CA3 neurons and the mossy cells of the subgranular area of the dentate gyrus [20]. In contrast, neuronal hGDH2 expression is not detected in the cerebellar cortex that lacks pyramidal neurons [20]. 

The regulatory properties acquired by hGDH2 during its evolution enable the recruitment of the enzyme upon intense excitatory firing, as noted above. These properties allow the enzyme to remain dormant under baseline conditions (displaying only 4–6% of its catalytic capacity) and be called into action during excitation. Specifically, because excitatory transmission requires high energy consumption, a process associated with increased conversion of ATP to ADP, the formed ADP can markedly activate hGDH2 (by 2400% at 1.0 mM ADP), as noted above. Thus, the ability of hGDH2 to hold in check substantial catalytic power releasable upon elevated energy demands permits robust glutamate metabolism upon bursts of glutamatergic firing required for long-term potentiation (LTP). Indeed, recent studies have revealed that *GLUD2* potentiates glutamatergic transmission in the hippocampus, resulting in enhanced synaptic plasticity, as described below. 

### 8.3. GLUD2 Enhances Synaptic Plasticity (LTP) by Increasing NMDA Receptor Currents

Electrophysiological experiments reveal that theta-burst-evoked long-term potentiation (LTP) is significantly enhanced in hippocampal CA3-CA1 synapses of Tg mice, compared to control mice [20]. Thus, following theta-burst stimulation, the synaptic response, recorded from CA1, is substantially more greatly enhanced in the Tg mice than in Wt mice. Patch-clamp recordings from CA1 pyramidal neurons showed that the amplitude, but not the frequency, of spontaneous excitatory postsynaptic currents (sEPSCs) is enhanced in Tg mice, suggesting that *GLUD2* increases basal glutamatergic synaptic transmission [20]. Additionally, the *GLUD2* Tg mice exhibited an enhanced frequency and amplitude of spontaneous NMDA receptor currents [20]. These results suggest that *GLUD2* potentiates synaptic plasticity by enhancing excitatory pot-synaptic mechanisms, specifically the NMDA receptor’s function [20]

### 8.4. GLUD2 Enhancement of Synaptic Plasticity Is Lactate-Mediated

Because metabolic studies using [U-13 C] glutamate have demonstrated that astrocytes metabolize glutamate through the TCA cycle generating lactate [66,67], and because lactate is an important player in glia–neuron interaction mechanisms that lead to memory consolidation [65], the effect of D-lactate (the inactive isomer of L-lactate capable of blocking its metabolism) was tested in the *GLUD2* Tg model. The results revealed that D-lactate essentially abolishes the enhancement of LTP in Tg mice but has little effect on the LTP of Wt mice [20]. While this finding suggests that *GLUD2* promotes synaptic plasticity by augmenting glia–neuron metabolic communication mediated by lactate, the mechanisms involved remain to be further understood.

Given that *GLUD2* encodes an enzyme that plays a key role in the metabolism of the excitatory amino acid glutamate, the finding that lactate is essential for the enhancement of synaptic plasticity induced by this gene needs to be understood within the context of the glia–neuron metabolic interaction that takes place in excitatory synapses (Figure 3). Such metabolic interactions between astrocytes and neurons have been extensively studied over the past decades and have been the subject of excellent reviews [75,76]. 

Converging evidence supports the view that glutamate is the central player in these mechanisms. Thus, according to the currently accepted model, glutamate, released from nerve terminals during excitation, is removed from the synaptic cleft via uptake into the surrounding astrocytes [77,78,79,80], where it is largely converted to glutamine [75,76]. The produced glutamine is subsequently transported into the nerve endings to serve as a precursor of the transmitter glutamate, thus completing the “glutamine/glutamate cycle”. However, the “glutamine/glutamate cycle” is not stoichiometric [81], suggesting that part of the transmitter glutamate is not recycled. 

Indeed, labeling studies in astrocytes, exposed to 0.5 mM of extracellular glutamate, metabolize the amino acid through the TCA cycle [66,67,68,81,82]. For this, glutamate is transported into the mitochondria, where it is converted to α-ketoglutarate, predominantly via GDH. Subsequently, α-ketoglutarate is metabolized through the TCA cycle, giving rise to lactate in the cytoplasm [66]. Specifically, using [U-13 C] glutamate, Ewald et al. [66] found that a substantial part of the label is incorporated into lactate, with most of the synthesized lactate being released by the astrocytes. The role of oxidative deamination of glutamate in these processes is underscored by observations in GDH-deficient astrocytes showing that, in the absence of glucose, lactate production from glutamate is reduced [83]. The studies by Sonnewald et al. [66] were performed in the presence of relatively high extracellular glutamate concentrations (0.5 mM) (approaching those present in the synaptic cleft during excitation). Under these conditions, metabolism of glutamate in mitochondria proceeds predominantly via the GDH-catalyzed oxidative deamination pathway [81,82,84,85,86,87]. In contrast, lower extracellular glutamate levels (0.1–0.2 mM) favor the transamination pathway [88]. As such, GDH is essential for the handling of glutamate loads associated with intense excitatory firing and consequently with high synaptic glutamate levels [89].

**Figure 3 ijms-25-05297-f003:**
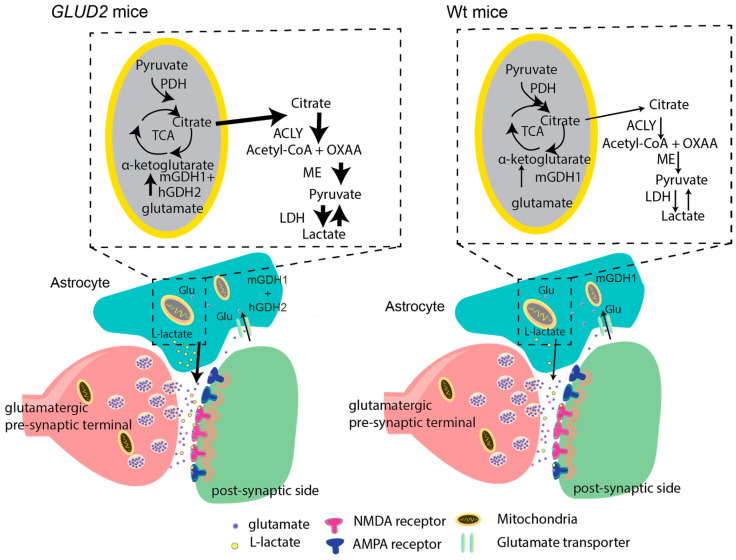
Schematic representation of a tripartite glutamatergic synapse in the hippocampi of transgenic (*GLUD2*) and wild-type (Wt) mice. Glutamate (GLU), released from presynaptic nerve terminals during neurotransmission, acts on post-synaptic NMDA and AMPA receptors. Synaptic glutamate is then rapidly removed from the synaptic cleft by uptake into the surrounding astrocytes (small arrow), where it is in part transported into the mitochondria. In the Wt mice, glutamate is converted to α-ketoglutarate via the endogenous mGDH1, whereas in the *GLUD2* Tg mice, this reaction is catalyzed by both the expressed hGDH2 and the mGDH1. α-ketoglutarate is subsequently metabolized through the TCA cycle, giving rise to lactate in the cytoplasm [66]. Indeed, metabolic studies using [U-13 C] glutamate have shown substantial incorporation of the label into lactate in a pattern that could only arise via metabolism of [U-13C] glutamate through the GDH-TCA cycle pathway [66]. Also, the observed labeling pattern of TCA cycle intermediates, such as citrate, permits the conclusion that part of citrate is exported to the cytoplasm, where it is catabolized by the ATP citrate lyase (ACLY) to oxaloacetate (OXAA) and acetyl-CoA [66]. Subsequently, OXAA is converted via the cytosolic malic enzyme (ME) to pyruvate, which gives rise to lactate by the action of LDH. Upon neuronal excitation, astrocytes release increased amounts of lactate (large arrow), which facilitates NMDA receptor signaling. These lactate-mediated effects are enhanced in the *GLUD2* Tg mice, leading to increased synaptic plasticity and synaptogenesis [20].

The findings that *GLUD2* enhances LTP through a lactate-dependent mechanism are consistent with observations showing that synaptic lactate potentiates NMDA receptor signaling [90,91]. In addition, lactate regulates the expression of genes involved in synaptic plasticity [75,90,91]. Of particular importance are data showing that lactate is essential for memory consolidation [65,92,93], a process associated with increased transport of lactate from astrocytes to neurons [65]. Although the role(s) of lactate in excitatory transmission, neuronal plasticity, and cognitive processes need to be better understood [75], the hypothesis that *GLUD2* advanced brain evolution by enhancing a lactate-dependent glial–neuron interaction [20] is consistent with recent comparative transcriptomic analyses, revealing that human brain evolution targeted pathways linked to neuronal and glial communication [8,9].

### 8.5. GLUD2 Enhances Dendritic Spine Density/Synaptogenesis

Because *GLUD2* enhances excitatory transmission and synaptic plasticity and because dendritic spines harbor most excitatory synapses in the brain [94], the density of the dendritic spines was evaluated in the *GLUD2* Tg model [20]. The results revealed that dendritic spine density is significantly increased in the hippocampus of Tg mice as compared to Wt animals. This increase involves both the mature dendritic spines (thin and mushroom) and the immature (stubby) dendritic spines [20]. Electron microscopy further revealed an increased number of synapses in the hippocampus of the Tg mice as compared to the Wt animals [20]. Hence, by enhancing synaptogenesis and dendritic spine density, *GLUD2* may promote the creation of new neuronal connections that constitute the structural basis for long-term memory. Moreover, the advent of *GLUD2* may have contributed to some of the cytoarchitectural features that distinguish the human brain from the chimpanzee brain, including a higher density of inter-neuronal connections and of temporal lobe white matter [95,96]. Also, because the evolution of excitatory projection neurons (rather than inhibitory neurons) is characteristic of the human brain [8,9], and because single-cell RNA expression data [60,61] have revealed that *GLUD2* expression is three-fold higher in excitatory neurons as compared to inhibitory neurons, the novel human gene may have contributed to the development of the unique cytoarchitectural features of the human brain. As such, the role that *GLUD2* has played in the emergence of human cognitive abilities is an exciting aspect that remains to be further understood.

### 8.6. GLUD2 Enhances Sensory Perception and Complex Cognition

As synaptic glutamatergic mechanisms play an important role in sensory functions, Tg and Wt mice were evaluated for their sensitivity to thermally induced pain and environmental illumination. Pain sensitivity was tested by using the “Hot Plate” and the Hargreaves method. Both tests reveal that Tg mice are significantly more sensitive to painful stimuli than Wt animals (*t*-test *p* < 0.001) [20]. Also, in the Light/Dark test, Tg mice proved significantly more sensitive to environmental illumination than their Wt littermates (*t*-test, *p* = 0.001). In addition, the Elevated Plus Maze test revealed that, compared to the Wt mice, the Tg animals prefer a closed chamber over an open space (*t*-test, *p* = 0.003). These behavioral changes likely reflect the strengthening of the innate rodent survival responses, probably brought about by *GLUD2*-induced enhancement of synaptic plasticity [20].

To test the effect of *GLUD2* expression on cognition, a battery of behavioral tasks, designed to assess various forms of cognitive function, was employed [20]. These included relatively simple cognitive tasks, such as the Novel Object Recognition Task (Recognition Memory), the Novel Object Location Task (Spatial Memory), and the Left–Right Discrimination task (Reference Memory), as well as more elaborate tasks, such as the Attentional Set-Shifting task (AS-ST), which assesses the ability of mice to adapt to changing external demands and which is considered a form of behavioral flexibility [97,98,99], and the Contextual Fear Conditioning/Extinction (CFC/E) task, which represents a form of inhibitory learning [100,101,102]. 

The results revealed that Tg and Wt mice performed equally well on the Novel Object Recognition Task, the Novel Object Location Task, and the Left–Right Discrimination task, with no significant differences found between the two groups [20]. On the other hand, the Tg mice performed better than Wt mice on certain phases of the AS-ST. This task comprises several phases that examine the ability of the animals to dig through bowls under changing conditions (bedding and/or of olfactory cues) in order to reap their reward. Results revealed that, in the “compound discrimination” phase (the animals need to differentiate between two different beddings in the presence of two different olfactory cues), Tg and WT mice performed similarly. However, in the “compound discrimination reversal” phase (the bedding and olfactory cues used are the same as for the “compound discrimination”, but the rewarded bedding is switched), the Tg mice performed significantly better than Wt mice (*p* = 0.01). In addition, a trend towards improved performance was detected for Tg mice (*t*-test, *p* = 0.06) in the “simple discrimination” phase (the animals need to differentiate between two different beddings). These results suggest that *GLUD2* expression enhances the animals’ ability to learn new strategies by inhibiting previously learned experiences [97,98,99].

The CFC/E task assesses the ability of the animal to encode and retrieve new memories that allow them to extinguish fearful responses developed during conditioning training [100,101,102]. The animals are initially placed in the conditioning chamber (context), where they receive a single electric shock. Then, upon returning to the context on each of the following 5 days, the mice are observed for possible episodes of freezing behavior (characteristic of fear). The results revealed that the Tg mice showed enhanced contextual fear extinction, as they froze significantly less often than Wt animals on the 5th day of testing (*t*-test, *p* = 0.03) [20]. The improved performance of the Tg mice in the CFC/E task suggests that *GLUD2* helps these animals acquire new reassuring memories while erasing previously felt fearful experiences. 

These findings, showing that transgenic expression of the human *GLUD2* gene promotes synaptic spine formation and synaptogenesis in the hippocampus in association with improved performance in complex cognitive tasks, are congruent with results of recent studies revealing a correlation between dendritic spine density and the animals’ performance in cognitive tests such as the attention set-shifting and fear extinction tasks [103,104]. 

Taken together, these data support the view that the novel human gene may have played an important role in human brain evolution by enhancing glutamatergic transmission, synaptic plasticity, synaptogenesis, and inter-neuronal connectivity, processes crucial for complex cognition. Also, the putative role of *GLUD2* in the evolution of excitatory neurons of the human brain, as noted here, is an exciting possibility that deserves to be further explored.

## 9. Conclusions 

Besides its large size and unique organization, the human brain is characterized by high rates of synaptic (mainly glutamatergic) transmission in association with elevated expression of synaptic proteins without changes in their structure. Recently, high-output single-cell profiling of the transcriptome, proteome, and epigenome of the human brain has revealed that evolution has induced changes in nerve cells that allow the creation of distinct circuitry [3,4], which is the structural basis of cognition [5,7]. However, the evolutionary origin of these adaptations remains poorly understood.

While synaptic transmission is conserved in primate lineages, a novel gene (*GLUD2*) that concerns glutamate metabolism emerged in the common ancestor of humans and apes and evolved along with increasing brain size and complexity. Following its emergence in the hominoid, *GLUD2* underwent rapid evolutionary adaptation that enabled enhanced enzyme function upon intense excitatory transmission, a process crucial to cognitive processes. In addition, *GLUD2* evolution permitted its expression in astrocytes and in pyramidal neurons of the human cerebral cortex. In these neurons, *GLUD2* is expressed in mitochondria distributed in the perikaryon and in the nerve terminals. These observations support the concept that the novel gene is involved in glutamatergic excitatory mechanisms that underlie cognitive functions. 

The possibility that the relatively recent emergence and evolution of *GLUD2* contributed to human cognitive advancement received support from the results of recent studies on the hippocampi of mice carrying the novel human gene [20]. The mice studied were transgenic for a region of the human genome (in the X chromosome) that contains the *GLUD2* gene and its regulatory elements. The transgenic (Tg) animals expressed hGDH2 in their brain in a pattern that closely resembled that of the human brain. This included expression in cortical pyramidal neurons and astrocytes throughout the neuropil. In the hippocampus, hGDH2 is expressed in the pyramidal cells of the stratum pyramidale (CA1-CA3) and the mossy hilar neurons of the dentate gyrus [20]. Both neuronal types are thought to play a crucial role in the cognitive process. 

Electrophysiological studies on hippocampal slices from Tg mice reveal that *GLUD2* significantly enhances long-term potentiation (LTP) in the CA3-CA1 synapses. Additional experiments using the patch-clamp technique reveal significant increases in the amplitude of sEPSC without changes in its frequency, findings implying that presynaptic activity is not significantly altered in a Tg hippocampus. Instead, the novel human gene enhances excitatory post-synaptic mechanisms by increasing the frequency and amplitude of the sNMDA currents. Importantly, the long-term strengthening of synaptic transmission, detected in the Tg hippocampus, is associated with increased structural synaptic plasticity (dendritic spine density and synaptogenesis), suggesting that *GLUD2* enhances inter-neuronal connectivity.

Disruption of lactate metabolism by D-lactate markedly attenuates the enhanced LTP in Tg animals, suggesting that *GLUD2* expression leads to increased synaptic plasticity by increasing the lactate-mediated metabolic coupling between astrocytes and neurons. Such lactate-dependent glia–neuron interaction potentiates NMDA receptor signaling and synaptic plasticity (as found in the hippocampi of the Tg mice) and is central to cognitive functions [90]. The concept that *GLUD2* has contributed to human brain evolution by enhancing astrocyte–neuron interaction is congruent with high-throughput single-cell comparative transcriptomic data showing that evolution has targeted pathways involved in neuron/glial communication [8,9]. 

The central argument, deriving from these observations, is that *GLUD2*, an enzyme of glutamate metabolism, enhances synaptic plasticity (LTP) through lactate-dependent potentiation of post-synaptic excitatory mechanisms, rather than by promoting pre-synaptic glutamatergic action. Regarding the mechanism(s) involved, *GLUD2* is thought to promote glial–neuron communication by enhancing the conversion of glutamate to lactate. This possibility is in accordance with metabolic labeling studies [66,67] showing that astrocytes metabolize extracellular glutamate through the GDH-TCA cycle pathway, thereby generating lactate [20]. 

Behavioral studies revealed that Tg mice exhibited enhanced sensitivity to pain and avoidance of open spaces and environmental illumination. The Tg mice also showed improved performance in aspects of attention set-shifting and contextual fear extinction, tasks thought to depend on higher cortical functions. 

These data demonstrate that a genetically determined post-synaptic excitatory potentiation, achieved by *GLUD2* expression, acts as a driving force for the long-term strengthening of excitatory synapses and the formation of inter-neuronal connections that subserve cognitive abilities. As these data link gene evolution with cortical excitatory mechanisms that mediate experience-dependent synaptic plasticity (a process shaped by environmental influences), the novel gene may contribute to a synergy between nature and nurture, mechanisms of fundamental importance for human evolution. Additional studies are needed to test these exciting possibilities and further evaluate the putative role of the novel human gene in human brain maturation and aging.

## Figures and Tables

**Figure 1 ijms-25-05297-f001:**
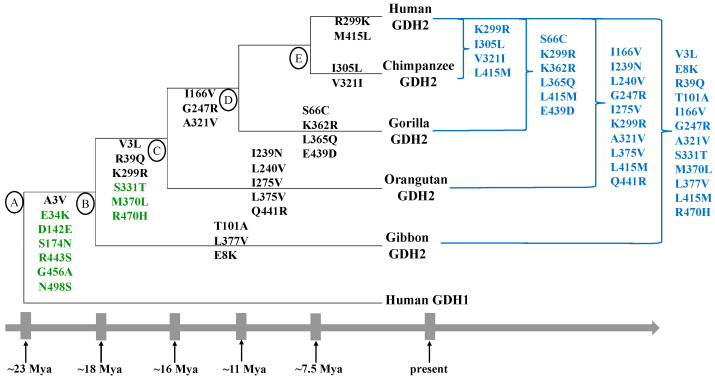
Divergence of the *GLUD2* gene among members of the hominoid radiation. Phylogenetic tree (adapted from references [22] and [34]) showing *GLUD2* evolution in primates. It has been estimated that *GLUD2* emerged about −23 Mya and evolved along the human and ape lineages. Depicted here are amino acid substitutions that occurred on the different branches (A,B, B,C, C,D, and D,E) of the phylogenetic tree. Amino acid replacements thought to have evolved under positive selection are green-colored, whereas those that differ between the GDH2 of the human and the GDH2 of non-human apes are blue-colored. Besides the amino acid replacements, shown here to have occurred in the gibbon lineage after its separation from that of the human and great apes, additional changes have been recently detected in Hylobates moloch, Symphalangus, and Nomascus [33]. Divergence times in millions of years (Mya) are shown at the bottom of the figure (not to scale). These time estimates are based on References [35,36,37,38,39]. In contrast to *GLUD2*, its ancestor, the *GLUD1* gene, remained conserved [22].

**Figure 2 ijms-25-05297-f002:**
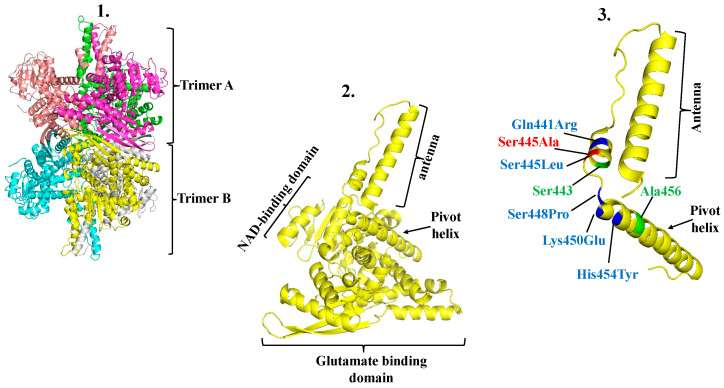
Structural model of human GDH2. Graphic representation of the 3D structure of the hGDH2 hexamer, comprising two trimers (**1A**,**B**) (PDB code: 6G2U). Each color corresponds to one of the six identical subunits. In (**2**), one monomer is depicted. The NAD+-binding domain, the glutamate-binding domain, the antenna, and the pivot helix are identified. In (**3**), the precise locations of residues (Ser-443 and Ala-456) that provide hGDH2 with unique properties are shown in green. Ser-443 is located in the small C-terminal α-helix of the antenna and Ala-456 is in the pivot helix. Also, the precise location of the hGDH2 mutations Glu441Arg, Ser445Leu, Ser445Ala, Ser448Pro, Lys450Glu, and Hist454Tyr that affect the basal activity and/or regulation (see text and Ref [35]) are shown in blue, except for the Ser445Ala change, which modifies Parkinson’s disease onset. This is shown in red. Interestingly, except for Ser445Ala, all these mutations occur in hGDH1, attenuating GTP inhibition and causing the HI/HA syndrome. These observations imply that the same mutations in hGDH1 and hGDH2 can have diverse functional consequences. The PyMOL Molecular Graphics System, Version 2.5 Schrödinger, LLC was used to create the graphics. This was obtained from Schrödinger, Inc., New York, NY, USA.

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
