# Peer review of "Evolution of Glutamate Metabolism via GLUD2 Enhances Lactate-Dependent Synaptic Plasticity and Complex Cognition"

_ijms, 2024, doi:10.3390/ijms25105297_

Round 1

Reviewer 1 Report

Comments and Suggestions for Authors

Plaitkis et al. review paper on GLUD2 gene and its role in cellular energetics, lactate metabolism, and brain Biology is very interesting topic. The authors cited relevant papers related to GLUD2 in cellular biology metabolism and brain biology and its role in brain biology. There were minor typographical errors in the font of the legend for the line 361 - the legend for the figure number 3 should be same as other figures legend in small font. The authors should have cited the graphs taken from other publications references in legend of the figures.

The review article flows well. They discussed the function of GLUD2 in brain biology and lactate metabolism, together with its biological genesis and evolutionary history among common ancestors. The paper's references are pertinent; every effort was made to include the most recent publications that were cited inside the framework. Figures from published papers were used to highlight the significance of GLUD2, a protein found in mitochondria that recycles glutamate during neurotransmission, as well as the most recent findings from the journals.

Author Response

We thank the Reviewer for evaluating our manuscript.

Below please find a point-by-point response to his comments and suggestions:

“There were minor typographical errors in the font of the legend for the line 361 - the legend for the figure number 3 should be same as other figures legend in small font. The authors should have cited the graphs taken from other publications references in legend of the figures”.

The original Figure 3 and 4 were omitted as suggested by Reviewer 2. 

We would like to thank again the reviewer for the careful evaluation of our manuscript. We hope that you will find the extensively revised manuscript acceptable for publication in the International Journal of Molecular Science.

Reviewer 2 Report

Comments and Suggestions for Authors

The paper by Andreas Plaitakis et al. reviews the current advances in understanding of GLUD2 brain function in the context of the brain evolution and metabolism. Although there are certain advances in the field, the topic of evolution itself is not so well presented. Visual representation in this work is either of bad quality (Figs 1,2,5) or copied from a previous paper with slight modifications (Figs 3,4). Starting from page 7 (see details below), the amount of text duplicated with other publications greatly increases resulting in more than 30% overall text similarity, excluding bibliography. As I've seen before, this parameter is normally much lower. The duplicated text can be noticed while reading the paper and going into the details in the references. Altogether, it spoils perception of the paper. Such duplicating parts should be shortened, with proper citations used. Moderate editing of English is also needed.

More detailed comments are provided below.

Since the paper title is "Evolution of Glutamate Metabolism via GLUD2 Enhances Lactate-Dependent Synaptic Plasticity and Complex Cognition", a few sentences about 1) Lactate-Dependent Synaptic Plasticity and 2) GLUD2 as a protein of glutamate metabolism are necessary for the Introduction. A clearly stated aim of this review is also needed in the Introduction too.

Can you prove or rewrite the sentence in lines 100-101 "However, the new gene evolved independently on the lines that descendent to great apes."? Additionally, the great apes do not include gibbons, please correct. Plus, "lineages" instead of "lines" should be used.

Line 101 - "As a result, GLUD2 diverged significantly" and further. Clarify the significance test or use another word.

Lines 102-104 and Fig. 1. Such evaluation is rather strange, because normally, the distance from the common ancestor is compared. Please correct. The species should be fully mentioned. Abbreviation "AGM" is not deciphered in the figure caption. How do you count the deletion observed in gibbon?

Please, also add a citation to the following statement: "Phylogenetic evidence suggests that the human neocortex expanded rather rapidly about -2Mya"

Line 107-109. There are papers, including yours, on hGDH mutants. Corresponding mutants could be mentioned here.

Lines 114-116. The text duplicates an already mentioned sentence.

Lines 116-123. These sentences should be presented as an illustration rather than it is listed in the text here. Also use the comment about the deletion in gibbon.

Lines 144-146 - the sentence is misleading and should be corrected.

Line 151-152. This sentence is not informative. Either add information about the differences, or delete the sentence.

Fig. 2. The hexamer structure is not correct from the biological point of view. It would be great if you could correct the biological assembly of your structure in the PDB as well, but at least in the paper you should use a proper image of the hexamer, which is a dimer of trimers. Currently, there is no use in Fig. 2 for the manuscript, so it can be omitted.

Further reading suggests, the figures 3 and 4 are in fact modified copies from a recently published paper (31). The need for such a duplication of already published data is not clear. It is not a combination/comparison of data from several papers, thus the original paper could be just cited.

Figure 5 is not clearly described. Moreover, from the text it can be suggested, that it is not complete: "Released glutamate is then rapidly removed from the synaptic cleft by uptake (arrows) into the surrounding astrocytes." For example, there are no mentioned arrows in the figure. Visualization of presumably mitochondria and Glu-Lactate conversion is not clear.

Moreover, analysis of the text used now by MDPI identified more than 30% of the text to be similar to the previous papers (excluding references and quotes). Particularly, 10% of the text is similar to the reference (31). Sometimes, paragraphs or big blocks of text are identical, for example in pages 7, 10, 11, 14-17. Such duplication can be observed during reading too, leading to a bad feeling about the paper and the novelty of the text, although its scientific details are interesting, and the authors are specialists in the field.

Minor comments:
Lines 84,88 - Emergence would likely suit better than "Birth" for a gene

Line 95 - ref (https://doi.org/10.1371/journal.pgen.1000150) would fit here too. 

Line 138 - "roads" - correct.

Line 180- "lower affinity" compared to what?

Comments on the Quality of English Language

There are some sentences, which need a correction. For example, the word "lines" instead of "lineages" is used. A sentence starting with "These have largely elucidated" should be corrected, as well as the sentence in line 168. The words "birth", "significance", "independently"... should be used more carefully. 

Author Response

We thank the Reviewer for evaluating our manuscript and for his constructive comments and suggestions. Based on his recommendations, we have extensively rewritten and revised our paper, and we are pleased to resubmit it for consideration to be published in the International Journal of Molecular Science.

We have reduced the self-citation references to about 15% of the total references by eliminating some of our self-citations and by increasing the total number of references.

Below please find a point-by-point response to the reviewer's comments and suggestions.

  1. Starting from page 7 (see details below), the amount of text duplicated with other publications greatly increases resulting in more than 30% overall text similarity, excluding bibliography. As I've seen before, this parameter is normally much lower. The duplicated text can be noticed while reading the paper and going into the details in the references. Altogether, it spoils perception of the paper. Such duplicating parts should be shortened, with proper citations used”

Response: The comments and suggestions of the reviewer are well taken. We have accordingly revised and extensively rewritten parts of the manuscript thought to include material reduplicated from our previous publications. As suggested by the reviewer, such parts were shortened, and additional citations were provided.

  1. Since the paper title is "Evolution of Glutamate Metabolism via GLUD2 Enhances Lactate Dependent Synaptic Plasticity and Complex Cognition", a few sentences about 1) Lactate Dependent Synaptic Plasticity and 2) GLUD2 as a protein of glutamate metabolism are necessary for the Introduction. A clearly stated aim of this review is also needed in the Introduction too”.

Response: In the revised manuscript, a paragraph was added at the end of the Introduction (page 2, last paragraph), which reads as follows:

 “To this day, only a few new genes, thought to have played a role in human brain evolution, have been characterized [4]. Of these, GLUD2 is of particular importance, given that: a) it emerged in the hominoid ancestor and evolved under positive selection concomitantly with brain development; b) it encodes human glutamate dehydrogenase2 (hGDH2), an enzyme central to the metabolism of glutamate, the major excitatory transmitter involved in cognitive processes, and c) hGDH2 acquired unique functional properties that allow the novel enzyme to be called into action upon intense excitatory firing, a process required for long-term strengthening of synapses and the creation of inter-neuronal connections. The possibility that hGDH2 has contributed to the acquisition of traits unique to human has been supported by recent investigations in mice transgenic for the human GLUD2, showing that the human gene enhances synaptic plasticity/synaptogenesis and complex cognition. Moreover, enhancement of synaptic plasticity by GLUD2 is lactate-mediated, thus providing additional evidence that synaptic lactate mechanisms are essential to memory consolidation. GLUD2 has also adapted to the particular metabolic needs of non-neural tissues where expressed. This review aims at detailing these advances and their implications for understanding the role of GLUD2 in human biology”.

  1. Can you prove or rewrite the sentence in lines 100-101 "However, the new gene evolved independently on the lines that descendent to great apes."? Additionally, the great apes do not include gibbons, please correct. Plus, "lineages" instead of "lines" should be used. Line 101 - "As a result, GLUD2 diverged significantly" and further. Clarify the significance test or use another word.

Response: We have rewritten these lines as suggested by the reviewer. Specifically, when referring to the group of species that includes the gibbons we use the term apes, instead of great apes. We also use lineages inserted of lines. Significance was determined by the Pearsons’ correlation as indicated in the first paragraph of page 5.  

  1. Lines 102-104 and Fig. 1. Such evaluation is rather strange, because normally, the distance from the common ancestor is compared. Please correct. The species should be fully mentioned. Abbreviation "AGM" is not deciphered in the figure caption. How do you count the deletion observed in gibbon?

Response: We have re-written these sentences (page 5, first paragraph): The new text reads as follows: “Thus, as shown in Figure 1, the human GDH2 differs from the chimpanzee GDH2 by 4 amino acid residues, from the gorilla GDH2 by 6, from the orangutan GDH2 by 10, and from the gibbon GDH2 by 12. These amino acid differences correlate significantly (Pearson correlation r=0.9957; p=0.004) with the time distances (in Mya) between the human and each ape and their last common ancestor(Figure 1)”.

  1. Please, also add a citation to the following statement: "Phylogenetic evidence suggests that the human neocortex expanded rather rapidly about -2Mya" Line 107-109.

Response: Citation (Ref 41) was added.

  1. There are papers, including yours, on hGDH mutants. Corresponding mutants could be mentioned here. Lines 114-116. The text duplicates an already mentioned sentence.

Response: In the revised manuscript, the duplicated text has been deleted. Also, GDH mutants, studied by us and others, have been mentioned on page 3 and 7. We also refer to the evolutionary mutations in the MTS as follows (page 4; second paragraph): “In addition to these changes in the mature GDH2, evolutionary replacements also occurred in the mitochondrial targeting sequence (MTS) of the protein. These conferred an enhanced mitochondrial targeting capacity [32,40]. Two positive selected evolutionary amino acid substitutions that occurred in the hominoid are thought to provide this novel property to GDH2. Specifically, the Glu7Lys evolutionary change, which is conserved among apes and which replaces a negatively charged residue (Glu) with a positively charged one (Lys) may play a key role in enhancing the transport of the GDH2 protein into the mitochondria [32]. Also, the Asp25His change, which replaces a negatively charged residue (Asp) with a partially positively charged residue (His), may have also contributed to the mitochondrial targeting adaptation of hGDH2 [32]. In the gibbon lineage, however, a three amino acid deletion (residues: 24, 25 and 26) that includes the Asp25His residue, reduces the MTS enhanced targeting capacity [32].”

  1. Lines 116-123. These sentences should be presented as an illustration rather than it is listed in the text here. Also use he comment about the deletion in gibbon.

Response: As suggested by the reviewer, the evolutionary amino acid replacements that differ between the human GDH2 and the GDH2 of each ape are illustrated in the revised Figure 1. The gibbon deletion is described above.

  1. Lines 144-146 - the sentence is misleading and should be corrected. Line 151-152. This sentence is not informative. Either add information about the differences, or delete the sentence.

Response: These lines were re-written as follows (page 6; last paragraph): “Additionally, the crystallization of hGDH2 allowed in-silico studies on the structural evolution of the primate protein [34]. These studies, using AlphaFold, examined changes in the GDH2 structure occurring during the evolutionary transition from extinct primate ancestors to modern apes, including humans. They revealed that the initial seven evolutionary amino acids, which occurred shortly after the retro-transposition event as detailed above (branch A-B in Figure 1), served as a basis for subsequent modifications that fine-tuned its enzymatic properties”.

  1. Fig. 2. The hexamer structure is not correct from the biological point of view. It would be great if you could correct the biological assembly of your structure in the PDB as well, but at least in the paper, you should use a proper image of the hexamer, which is a dimer of trimers. Currently, there is no use in Fig. 2 for the manuscript, so it can be omitted.

Response: In the revised manuscript, Figure 2 has been substantially modified. The new Figure 2 shows: 1.) the hGDH2 hexamer made up of two trimers (Figure 1: A and B); 2.) one hGDH2 monomer displaying its functional domains, and 3.) the antenna and pivot helix structures, in which the precise location of amino acid substitutions involved in hGDH2 evolution is depicted. Also, the location of mutations in the antenna and pivot helix that affect hGDH2 activity is indicated, including a naturally occurring variant (Ser1445Ala) that modifies Parkinson’s disease onset (Ref. 55).

  1. Further reading suggests, the figures 3 and 4 are in fact modified copies from a recently published paper (31). The need for such a duplication of already published data is not clear. It is not a combination/comparison of data from several papers, thus the original paper could be just cited.

Response: Figures 3 and 4 have been deleted as suggested by the reviewer.

  1. Figure 5 is not clearly described. Moreover, from the text it can be suggested, that it is not complete: "Released glutamate is then rapidly removed from the synaptic cleft by uptake (arrows) into the surrounding astrocytes." For example, there are no mentioned arrows in the figure. Visualization of presumably mitochondria and Glu-Lactate conversion is not clear.

Response: The original Figure 5 (present Figure 3) has been revised as suggested by the reviewer. Specifically, the figure depicts an arrow indicating glutamate uptake (through a specialized glutamate transporter located in the cytoplasmic membrane of the astrocyte) and another arrow showing lactate export from the astrocyte. Astrocytic mitochondria are visualized, along and the metabolic pathways involved in the conversion of glutamate to lactate.

  1. Moreover, analysis of the text used now by MDPI identified more than 30% of the text to be similar to the previous papers (excluding references and quotes). Particularly, 10% of the text is similar to the reference (31). Sometimes, paragraphs or big blocks of text are identical, for example in pages 7, 10, 11, 14-17. Such duplication can be observed during reading too, leading to a bad feeling about the paper and the novelty of the text, although its scientific details are interesting, and the authors are specialists in the field.

Response: As noted above, we have extensively rewritten parts of the manuscript thought to include reduplicated material from our previous publications.

Regarding the minor comments of Reviewer 2:

Lines 84,88 - Emergence would likely suit better than "Birth" for a gene Line 95 - ref (https://doi.org/10.1371/journal.pgen.1000150) would fit here too. Line 138 - "roads" - correct. Line

As suggested, we used “emergence” instead of “birth”. Also, the sentence that includes “roads” has been rephrased.

180- "lower affinity" compared to what?  

Here the original text reads: While GTP potently inhibits (IC50 0.1-0.2 175 μM) hGDH1, ADP activates the enzyme with a 100-200-fold lower affinity (AD50 ~18-20 μM)”. As such, the ADP affinity is compared to that of GTP. However, to make this point clearer, we revised the text as follows: “While GTP potently inhibits (IC50 0.1-0.2 μM) hGDH1, ADP activates the enzyme (AD50 ~18-20 μM), with the two allosteric effectors acting antagonistically”.

Comments on the Quality of English Language There are some sentences, which need a correction. For example, the word "lines" instead of "lineages" is used. A sentence starting with "These have largely elucidated" should be corrected, as well as the sentence in line 168. The words "birth", "significance", "independently"... should be used more carefully.

In the revised manuscript, these sentences have been corrected and/or rephrased.

We would like to thank again the reviewer for the careful evaluation of our manuscript and his constructive suggestions.

We hope that he will find the extensively revised manuscript acceptable for publication in the International Journal of Molecular Science.

Round 2

Reviewer 2 Report

Comments and Suggestions for Authors

The new version of the manuscript shows a very good example of a revision, and the new version provided by Andreas Plaitakis and co-authors can be read with pleasure.

First, I'd like to add a comment to the figures. The current Fig. 3 is a bit difficult to read, and the choice of black background for the mitochondria is very bad. As a result it is difficult to see the difference between the two parts of the figure. A light gray background and black font would be a better choice here. Additionally, could you enlarge the font and make the difference between arrows in the two parts of the picture more evident?

The font of Figure 2 should be enlarged too. The order of numbers (1-3) from the right to the left should be corrected to a more common variant (from the left to the right).

The existence of the reverse mutations found in a gibbon sounds very interesting, because, as I understand, it also raises a question whether the positive selection of these residues is significant or not. I went deeper into the first paper claiming the positive selection (10.1038/ng1431) and found, that "The analysis also identified an excess of nonsynonymous (KA) over synonymous substitutions (KS) per site (KA/KS > 1) for the branches after the duplication event leading to humans and the great apes, indicative of accelerated protein evolution driven by positive selection10; this excess was not statistically significant. But a maximum likelihood analysis that tests for selection at certain sites on these branches..."
The results of the maximum likelihood analysis indicate that "Amino acid changes E34K, D142E, S174N, R443S, G456A, N498S, S331T, M370L and R470H occurred at sites that are inferred to have been potentially under positive selection (P > 0.95)." As far as I understand, the mutations A3V, V3L and K299R are not listed, because their further mutations are available. Thus, the reverse mutations of R443S and G456A would likely question the proposed positive selection within these sites. As I can see, this is briefly mentioned in Ref [33] too.

Thus, although not questioning the positive selection of GDH2 in general, I would suggest being more accurate with the old references claiming existence of positive selection and probably add a comment about its insignificance (Ref 22) and current possible contradiction for the two individual residues (Ref 33).

Line 134. Symphalangus and Nomascus are not the individual members, these are genuses.

Comments on the Quality of English Language

A few phrases such as "retro-transportation" instead of retrotransposition (or retroposition) should be corrected.
A careful correction by the English Editor (and by the authors of course) would be required, if the Editor accepts the paper.

Author Response

We thank the reviewer for his thorough and in-depth evaluation of our manuscript, and for his insightful comments. His rigorous approach is helpful to our efforts to make our review as accurate and comprehensive as possible. We have revised our manuscript in response to the reviewer’s comments and suggestions and are pleased to re-submit it for consideration to be published in the International Journal of Molecular Sciences.

Regarding points raised by Reviewer

1, The current Fig. 3 is a bit difficult to read, and the choice of black background for the mitochondria is very bad. As a result it is difficult to see the difference between the two parts of the figure. A light gray background and black font would be a better choice here. Additionally, could you enlarge the font and make the difference between arrows in the two parts of the picture more evident?  The font of Figure 2 should be enlarged too. The order of numbers (1-3) from the right to the left should be corrected to a more common variant (from the left to the right).

Response:

Figure 2 and 3 were modified as suggested by the Reviewer.

Specifically, the font in Figure 2 was enlarged and the order of the numbers 1-3 was changed from left to right.

Similarly, the font was enlarged in Figure 3 and the arrows in the two parts of the pictured were modified to make the difference between the wild-type and the GLUD2 Tg astrocytes more evident. Also, the background for the mitochondria was changed to light gray as suggested by the reviewer.

  1. The existence of the reverse mutations found in a gibbon sounds very interesting, because, as I understand, it also raises a question whether the positive selection of these residues is significant or not. I went deeper into the first paper claiming the positive selection (10.1038/ng1431) and found, that "The analysis also identified an excess of nonsynonymous (KA) over synonymous substitutions (KS) per site (KA/KS > 1) for the branches after the duplication event leading to humans and the great apes, indicative of accelerated protein evolution driven by positive selection10; this excess was not statistically significant. But a maximum likelihood analysis that tests for selection at certain sites on these branches..."
    " The results of the maximum likelihood analysis indicate that "Amino acid changes E34K, D142E, S174N, R443S, G456A, N498S, S331T, M370L and R470H occurred at sites that are inferred to have been potentially under positive selection (P > 0.95)." As far as I understand, the mutations A3V, V3L and K299R are not listed, because their further mutations are available. Thus, the reverse mutations of R443S and G456A would likely question the proposed positive selection within these sites. As I can see, this is briefly mentioned in Ref [33] too. Thus, although not questioning the positive selection of GDH2 in general, I would suggest being more accurate with the old references claiming existence of positive selection and probably add a comment about its insignificance (Ref 22) and current possible cs ontradiction for the two individual residues (Ref 33).

Response

The skepticism expressed by the reviewer regarding the positive selection of GDH2 is well taken. It should however be noted that, in addition to the findings of Burki and Kaessmann (Ref 22), who used the “maximum likelihood” analysis to identify a subset of sites in the mature protein with KA/KS values >1.0 (p<0.01) indicative of positive selection, Rosso et al (Ref 32) also found a significant excess of nonsynonymous (KA) over synonymous (KS) substitutions (p<10-3) when their analysis was restricted to the first 159 nucleotides that code for the mitochondrial targeting sequence (MTS) of 53 amino acids.

The existence of the reverse mutations found in a gibbon sounds very interesting, because, as I understand, it also raises a question whether the positive selection of these residues is significant or not.

Response

The positive selection of these residues is reported by Burki and Kaessmann (Ref 22) to have occurred on the human and great ape lineages: “This might indicate that functional adaptation of GLUD2 driven by positive selection progressively occurred on the line leading to humans and great apes”. Hence, it is unclear whether the reversal of the Arg443Ser and Gly456Ala mutations in the Hylobates moloch (which may have followed a distinct evolutionary path) contradicts this assumption.  

Most intriguing is the absence of gibbon GLUD1 sequences in the UniProt (Ref 33), raising the theoretical possibility (according to Aleshina and Aleshin) that GLUD1 might have become a pseudogene after the emergence of GLUD2 in the gibbon. While this seems unlikely, reversal of the Arg443Ser and Gly456Ala mutations reinstate ancestral amino acids present in hGDH1 that contribute to the housekeeping properties of the enzyme. As such, whether this reversal permitted the GDH2 to take over some of the hGDH1 duties and whether this affected the evolution of hGDH1 remains to be further explored.  

Nevertheless, we do agree with the reviewer that reversal of Arg443Ser and Gly456Ser mutations in the gibbon raise important questions regarding the evolutionary trajectory of GLUD2 and our understating of the role of these residues in GDH2 adaptation. We have accordingly revised our manuscript, taking into consideration the current controversies regarding the implications of these reversals in GLUD2 evolution and function. Specifically, the last paragraph of page 3 reads as follows:

Phylogenic evidence suggests that the Arg443Ser and Gly456Ala changes occurred along with five additional evolutionary replacements (Ala3Val, Glu34Lys, Asp142Glu, Ser 174Asn, Asn498Ser) in the first few million years after the gene reduplication event (Figure 1; branch A-B). Six of these seven amino acid changes are present in every member of the hominoid radiation that possesses the GLUD2 gene (Figure 1). An exception is the Hylobates moloch genus of the gibbon family, in which the Arg443Ser and Gly456Ala changes were reversed in association with the appearance of four new mutations [33]. These observations have raised questions regarding the positive selection of these sites during the GLUD2 evolution, and the role of these residues in the functional adaptation of GDH2 [33]. Concerning the latter, it is presently unclear whether the four new amino acid replacements that emerged along with this reversal in Hylobates moloch provide to gibbon GDH2 properties similar to those conferred by the Arg443Ser and Gly456Ala replacements to human GDH2 [33]. Moreover, particularly intriguing is the absence of gibbon GLUD1 sequences in the UniProt [33], raising the theoretical possibility (according to Aleshina and Aleshin) that GLUD1 might have become a pseudogene after the emergence of GLUD2 in the gibbon. While this seems unlikely, reversal of the Arg443Ser and Gly457Ala mutations reinstates ancestral amino acids present in hGDH1 that contribute to the housekeeping properties of the enzyme. As such, whether this reversal permitted the gibbon GDH2 to take over some of the metabolic duties of hGDH1 and whether this has affected GLUD1 evolution remain to be further explored. Additional studies, including the functional characterization of gibbon GDH2s, are needed to better understand not only the evolutionary trajectory of the GLUD2 gene, but also its impact in the biology of these species”. 

  1. Line 134. Symphalangus and Nomascus are not the individual members, these are genuses.

We agree that the taxonomic rank for Hylobates moloch, Symphalangus and Nomascus is genera belonging to the gibbon family (Hylobatidae).

  1. A few phrases such as "retro-transportation" instead of retrotransposition (or retroposition) should be corrected.

These have been corrected.

  1. A careful correction by the English Editor (and by the authors of course) would be required, if the Editor accepts the paper.

We would be very glad to work with the English Editor of the Journal in order to improve the English language of our review.

Again, we would like to thank you and the reviewer for evaluating our manuscript and we hope that you will find the revised version suitable for publication in the International Journal of Molecular Sciences.